# CEPR2 perceives group II CEPs to regulate cell surface receptor-mediated immunity in Arabidopsis

Jakub Rzemieniewski[1], Patricia Zecua-Ramirez[2], Sebastian Schade[2], Zeynep Camgöz[1], Genc Haljiti[3], Sukhmannpreet Kaur[1], Christina Ludwig[3], Ralph Hückelhoven[1], Martin Stegmann [1,2]*

1 Chair of Phytopathology, TUM School of Life Sciences, Technical University of Munich, Freising, Germany, 2 Molecular Botany, Institute of Botany, Ulm University, Ulm, Germany, 3 Bavarian Center for Biomolecular Mass Spectrometry (BayBioMS), Technical University of Munich (TUM), Freising, Germany

* martin.stegmann@uni-ulm.de

## Abstract

Plant endogenous peptides are crucial for diverse aspects of plant physiology. Among them, C-TERMINALLY ENCODED PEPTIDEs (CEPs) have recently emerged as important regulators of plant growth and stress responses. CEPs are divided into two major subgroups: group I CEPs and the less studied group II CEPs. We recently demonstrated that group I CEPs coordinate cell surface receptor-mediated immunity with nitrogen status in *Arabidopsis thaliana* (hereafter Arabidopsis). To mount full group I CEP responsiveness, the three phylogenetically related CEP RECEPTOR 1 (CEPR1), CEPR2 and RECEPTOR-LIKE KINASE 7 (RLK7) are required. Here, we provide evidence that biotic stress induces expression of the group II CEP peptide *CEP14*. CEP14 and the related CEP13 and CEP15 trigger hallmark immune signalling outputs in a proline hydroxylation pattern-dependent manner in Arabidopsis. Genetic data indicate that group II CEP members contribute to cell surface receptor-mediated immunity against bacterial infection. We further show that group II CEP perception primarily depends on CEPR2. Our work provides new insights into CEP function during biotic stress and sheds new light on the complexity of sequence-divergent CEP signalling mediated by specific endogenous receptors.

## Author summary

To protect themselves against pathogens, plants have evolved a sophisticated and intricate immune system that allows pathogen detection and the activation of appropriate defense responses. There is increasing evidence that plant immunity is regulated by endogenous signaling molecules. For example, plants use a plethora of secreted peptides to fine-tune immune responses. Because of their functional similarity to animal cytokines, these peptides are referred to as

provided the original author and source are credited.

**Data availability statement:** All relevant data are in the manuscript and its supporting information files. The Skyline documents, as well as all mass spectrometric raw files measured in PRM mode, have been deposited to Panorama Public doi: 10.1074/mcp.RA117.000543 and can be accessed via https://panoramaweb.org/CEP14.url.

**Funding:** This work was supported by the Deutsche Forschungsgemeinschaft (grant STE2448/4-1; STE2448/4-2; TRR356 TP B09 to MS; INST 95/1436-1 FUGG to CL), the Technical University of Munich (MS, RH, and CL) and Ulm University (MS). The funders had no role in study design, data collection and analysis, decision to publish, or preparation of the manuscript.

**Competing interests:** The authors have declared that no competing interests exist.

phytocytokines. These peptides are often recognized by plant endogenous receptors at the cell surface to induce downstream signalling events that contribute to pathogen defense. Here we show that the pathogen-inducible peptide CEP14, together with the closely related CEP13 and CEP15, act as a phytocytokine in the model plant species *Arabidopsis thaliana*. We show that CEP13-CEP15 are perceived by CEPR2 to contribute to cell surface receptor-mediated resistance against the bacterial pathogen *Pseudomonas syringae* pv. *tomato*. Our study reveals a new endogenous peptide-receptor module that regulates effective immunity against bacterial infection in plants, providing potential applicability for future crop improvement.

## Introduction

Plants continuously monitor their extracellular environment for molecular signatures indicating potential threats. Microbe-associated molecular patterns (MAMPs) are conserved microbial components that can be recognized by pattern recognition receptors (PRRs) to activate pattern-triggered immunity (PTI). PRRs are either receptor-like kinases (RLKs) or receptor-like proteins (RLPs). RLKs possess kinase domains for signalling, while RLPs depend on RLK adaptors for signal relay [1]. Upon ligand binding, some PRRs form complexes with co-receptors belonging to the SOMATIC EMBRYOGENESIS RECEPTOR KINASES (SERKs) family, including BRASSINOSTEROID INSENSITIVE 1-ASSOCIATED RECEPTOR KINASE 1 (BAK1). Activated PRR complexes trigger downstream phosphorylation cascades involving receptor-like cytoplasmic kinases (RLCKs) which in turn relay the signal into the cell by, e.g., phosphorylating calcium ($Ca^{2+}$) channels to trigger cellular influx of $Ca^{2+}$ ions and initiating intracellular MITOGEN-ACTIVATED PROTEIN KINASE (MAPK) cascades. Further downstream events after MAMP perception include the biosynthesis of defence-related phytohormones such as ethylene and salicylic acid [1,2]. Plants tightly control their immune responses to ensure organismal homeostasis. In contrast, non-|physiological immune responses, such as those triggered by prolonged exposure of seedlings to MAMPs, can arrest growth [3].

Plant endogenous peptides have emerged as important regulators of cell surface receptor-mediated immunity, capable of either stimulating or inhibiting PTI. Referred to as phytocytokines, in analogy to metazoan systems, these peptides often exhibit multifunctional roles, influencing not only immune responses but also growth, development and abiotic stress adaptation [4–6]. This signalling versatility implies that phytocytokines can function as integrators of diverse exogenous and endogenous cues. Notably, many of these peptides can mimic MAMP-induced responses when applied exogenously, highlighting their potential to fine-tune defence pathways under various conditions [7–11].

Recently, we showed that C-TERMINALLY ENCODED PEPTIDEs (CEPs) are phytocytokines and can coordinate immunity with nitrogen (N) homeostasis [12]. CEPs are classified into two sequence divergent groups: group I CEPs

(CEP1-CEP12) and group II CEPs (CEP13-CEP15). Group I CEPs can be further divided into canonical CEPs (CEP1–3, CEP5-CEP11) and the non-canonical CEP4/CEP12, which differ in the positioning of key proline residues within the peptide domain. We recently demonstrated that group I CEPs trigger hallmark immune outputs and genetically contribute to disease resistance against *Pseudomonas syringae* pv. *tomato* (*Pto*). Canonical group I CEPs are perceived by CEP RECEPTOR 1 (CEPR1) and CEPR2, while the non-canonical CEP4 is additionally recognized by and binds to the phylogenetically related RECEPTOR-LIKE KINASE 7 (RLK7), which is essential for achieving full responsiveness and robust immunity [12–14]. Group II CEPs, on the other hand, remained poorly characterized.

In addition to their role as phytocytokines, class I CEPs have been extensively studied for their pivotal function in regulating root system architecture and adapting to N starvation [15]. By contrast, the functions of class II CEPs remain largely unexplored, with limited evidence suggesting their involvement in root development and potential roles in environmental stress adaptation [16,17].

Here, we provide evidence that the group II CEP *CEP14* is transcriptionally induced during biotic stress. Exogenous treatment with CEP14 and related group II CEP peptides induces hallmark immune responses in a proline hydroxylation-dependent manner and exhibits a unique receptor dependency distinct from other CEPs. We show that CEP13, CEP14 and CEP15 are primarily perceived by CEPR2 to activate immune signalling in Arabidopsis. We further demonstrate that group II CEPs are required for effective cell surface receptor-mediated immunity against *Pto*. A recent report has revealed that CEP14 regulates systemic acquired resistance (SAR) via CEPR2-BAK1/SERK4 complexes [18]. Our work not only supports the role of CEP14 in Arabidopsis immunity, but also uncovers additional functions of group II CEPs in modulating cell surface receptor-mediated immunity, thereby expanding the repertoire of CEP phytocytokines involved in regulating multiple layers of the plant defence system.

## Results

### Group II CEP members are expressed in different tissues and *CEP14* transcript abundance increases during biotic stress

We recently demonstrated that group I CEPs induce immune outputs and are required for basal resistance to *Pto* [12]. Surprisingly though, group I CEPs did not exhibit significant transcriptional changes in response to *Pto* infection or treatment with the 22 amino acid flagellin epitope flg22 [12]. Moreover, group I *CEPs* display tissue-specific expression patterns, with most members showing higher transcript levels in the roots [12].

In contrast, *CEP14* and *CEP15* showed a similar predicted expression strength in both above- and below-ground tissues, suggesting a broader distribution *in planta* (S1A Fig). *CEP13*, however, displayed a more restricted expression pattern, reminiscent of class I *CEPs*, with predicted localization primarily in the inflorescence (S1A Fig). Publicly available transcriptomic data revealed a notable upregulation of *CEP14* in response to two strains of *Pseudomonas syringae* and flg22 (S1B Fig). This differential expression was confirmed through RT-qPCR upon *Pto* DC3000 infection in adult leaves and flg22 treatment in seedlings (Fig 1A and 1B). Additional testing upon treatment with the 18 amino acid elicitor derived from bacterial ELONGATION FACTOR THERMO UNSTABLE (elf18) showed a significant increase in *CEP14* expression (Fig 1B). Previously published transcriptomic data confirmed this flg22 and elf18-mediated *CEP14* upregulation in seedlings (S1C Fig) [19]. Interestingly, *CEP14* transcripts were also induced in responses to other MAMPs, including pathogen-derived NECROSIS AND ETHYLENE-INDUCING PEPTIDE 20 (nlp20), a fragment of fungal cell wall (chitooctaose, CO8), as well as endogenous danger signals, including oligogalacturonides (OGs) and the endogenous PLANT ELICITOR PEPTIDE 1 (PEP1) phytocytokine (S1C Fig) [19].

The other group II CEPs showed less pronounced transcriptional regulation in response to defence-related stimuli. *CEP13* was weakly expressed in leaves and seedlings, with only a slight, non-significant increase following infection or MAMP treatments (Fig 1A and 1B). Consistent with publicly available RNA-seq data, *CEP15* expression was suppressed during infection with *Pto* DC3000 (Figs 1A and S1B), whereas MAMP treatments did not result in marked transcript

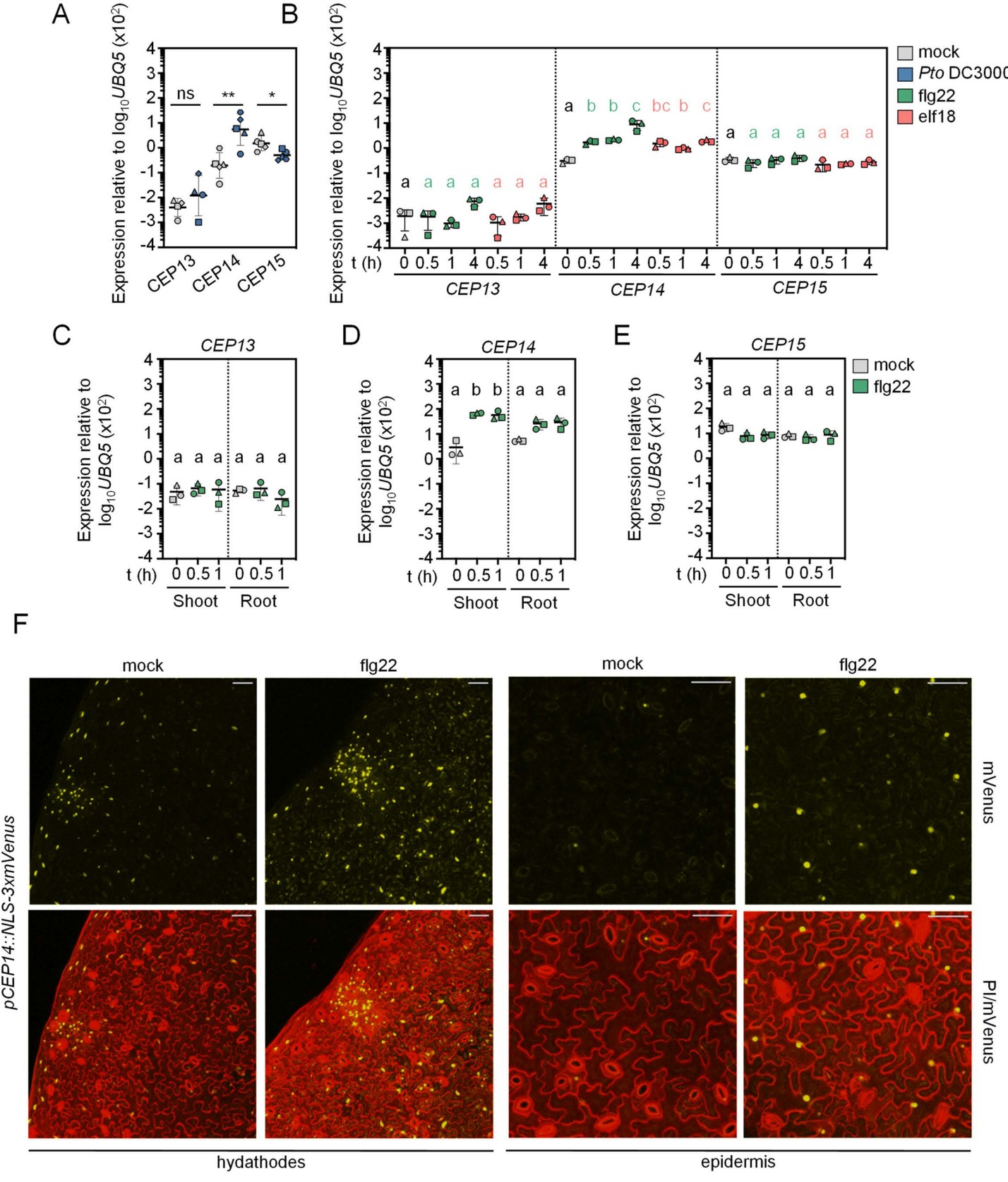

**Fig 1. The group II *CEP14* is transcriptionally induced during biotic stress.** A) RT-qPCR analysis of *CEP13, CEP14 and CEP15* expression in mock (ddH$_2$O) and *Pto* DC3000-inoculated leaves 24 h post-treatment. Housekeeping gene *UBQ5*; n = 4 (CEP13) and n = 5 (*CEP14, CEP15*) pooled from independent experiments, with mean ± SD. Due to deviations from normality, the statistical analysis was performed on log$_{10}$-transformed data (two-tailed Student's t-test, * p = 0.0168, ** p = 0.0042, ns = not significant). B) RT-qPCR analysis of *CEP* expression in Col-0 seedlings following flg22 (100 nM) or elf18 (100 nM) treatment for the indicated time. Housekeeping gene *UBQ5*; n = 3 pooled from independent experiments, with mean ± SD. Statistical comparisons for each *CEP* gene and treatment were performed separately. For the *CEP14* 0 h - flg22 comparison, data were log$_{10}$-transformed prior to analysis due to deviations from normality. Green letters indicate statistical comparisons between 0 h and flg22 treatments (one-way ANOVA, Tukey post-hoc test, a-b p < 0.0005, a-c p < 0.0001, b-c, p < 0.005), while coral letters indicate comparisons between 0 h and elf18 treatments (one-way ANOVA, Tukey post-hoc test, a-b p = 0.0003, a-bc, a-c p < 0.0001, b-c p = 0.0087). **C-E)** *CEP* expression in seedling shoots and roots. Col-0 seedlings were untreated (0 h) or treated with flg22 (100 nM) for the indicated time, after which shoots and roots were harvested separately for RT-qPCR analysis. Housekeeping gene *UBQ5*; n = 3 pooled from independent experiments, with mean ± SD. The dotted line indicates separate statistical comparisons for shoot and root *CEP* levels (one-way ANOVA, Tukey post-hoc test, a-b p < 0.01). F) Representative images of NLS-3xmVenus signal in *pCEP14::NLS-3xmVenus* lines upon mock (ddH$_2$O) or flg22 (1 µM) treatment for 16 h. The maximum projection of Z-stacks for mVenus is merged with the Z-stacked PI signal; scale bar = 50 µm. All experiments were performed in independent biological repeats at least three times with similar results.

changes (Figs 1B and S1C). Further analysis of flg22-treated seedling shoots and roots revealed expression patterns consistent with those observed in whole seedlings. *CEP13* transcript levels remained stable in both above- and below-ground tissues (Fig 1C). In contrast, *CEP14* was significantly upregulated in both tissues, with higher transcript increase in shoots than roots (Fig 1D). *CEP15* showed no significant expression changes in either roots or in shoots (Fig 1E).

To further resolve the spatial expression patterns of *CEP14*, we generated *pCEP14::NLS-3xmVenus* reporter lines. The fluorescent mVenus signal was specifically detected in the epidermis and mesophyll, with a particularly strong *CEP14* promoter activity observed at the hydathode region. Interestingly, no fluorescence signal was detected in guard cells or vascular tissues (Fig 1F), closely mirroring previous findings for the non-canonical group I *CEP4* promoter activity [12]. In line with RT-qPCR data, the *pCEP14::NLS-3xmVenus* line exhibited increased mVenus fluorescence upon flg22 treatment, particularly in epidermis and/or mesophyll cells (Fig 1F). These findings demonstrate that, unlike other *CEP* family members, *CEP14* expression is markedly upregulated during defence activation. Moreover, for all three group II CEP members, transcript accumulation was observed in both above- and below-ground tissues, despite the predicted inflorescence-specific expression of *CEP13*.

## Group II CEPs induce PTI-like responses in a proline hydroxylation pattern-dependent manner

Class II CEPs share an almost identical C-terminal VPSPG(V/I)GH sequence, which includes two conserved prolines, Pro9 and Pro11 [17] (S1D Fig). In contrast, the majority of canonical group I CEPs (CEP1–3, CEP5-CEP11) contain three conserved proline residues (Pro4, Pro7 and Pro11), while non-canonical group I CEPs share Pro4, but lack Pro7 and Pro11, instead having a proline at position 9 (Pro9) (S1D Fig). Apart from this variation, the overall peptide sequence of group I CEPs is largely conserved, including a characteristic C-terminal GXGH motif. Interestingly, group II CEPs feature the Pro9 residue typical of non-canonical group I CEPs (CEP4/CEP12), while also containing the Pro11 residue present in the rest of group I CEP family members. Although the C-terminal region of group II CEPs resembles that of the group I CEPs, their N-terminal sequences are markedly divergent. Since our previous study revealed that the perception of the non-canonical group I CEP4 requires RLK7, a described PAMP-INDUCED PEPTIDE (PIP) receptor, we included PIP sequences in our alignment (S1D Fig) [7,12]. Interestingly, the positioning of Pro9 and Pro11 in CEP13/14/15, along with specific N-terminal features, more closely resembles PIPs than canonical group I CEPs, suggesting a potential similarity between both phytocytokine families (S1D Fig).

To test whether group II CEPs can induce immune responses similar to those triggered by group I CEPs, we obtained the corresponding synthetic peptides. Given that Pro11 is present in most canonical class I CEPs and all identified CEP peptides to date exhibit hydroxylation at this position [14,20, 21], we hypothesized that Pro11 is the most likely residue to undergo hydroxylation *in planta*. In the case of PIPs, information about the identity of the *in vivo* accumulating peptide sequences is largely lacking, with the exception of TOLS2/PIPL3, which is hydroxylated at Pro9 [22]. We thus first

obtained CEP14 with Pro11 hydroxylated (CEP14HYP2). This peptide induced $Ca^{2+}$ influx in a Col-0 line expressing the $Ca^{2+}$ reporter Aequorin (Col-0AEQ) but failed to induce MPK3/6 activation, ethylene production and seedling growth inhibition (Figs 2B and S2A–S2C). To assess whether hydroxylation of Pro9 may affect bioactivity, we obtained synthetic peptides with hydroxylated Pro9 (CEP14HYP1) and double hydroxylated at Pro9/Pro11 (CEP14HYP1,2) (Fig 2A). The $Ca^{2+}$ influx triggered by both peptide versions was comparable to CEP14HYP2 (Fig 2B). Dose-dependent MPK3/6 phosphorylation assays showed that, unlike CEP14HYP2, CEP14HYP1 and CEP14HYP1,2 induced MAPK phosphorylation, with strong activity observed in the nanomolar range (Fig 2C). This data suggests that hydroxylation at Pro9 promotes CEP14 bioactivity.

**CEPR2 is required for CEP14 perception**

We next aimed to identify the CEP14 receptor. Our previous study has shown that group I CEP perception depends on CEPR1, CEPR2 and the phylogenetically related RLK7 [12]. We therefore tested whether these receptors are similarly required for CEP14HYP1,2-induced $Ca^{2+}$ influx. A newly generated cepr1AEQ single mutant line (CRISPR allele cepr1.2AEQ, S3A Fig) did not exhibit any differences in $Ca^{2+}$ influx compared to the wild type (S3B Fig). However, a novel cepr2AEQ line (CRISPR allele cepr2.2AEQ, S3B Fig) displayed strong $Ca^{2+}$ influx reduction in cepr2AEQ for all tested CEP14 peptide variants [12] (Fig 3A). The previously described rlk7AEQ line did not display defects in $Ca^{2+}$ influx triggered by CEP14 peptide variants, but the low residual activity in cepr2AEQ was further reduced in a new cepr2 rlk7AEQ double mutant (c2/r7AEQ, Figs 3A and S3D). This was particularly pronounced for CEP14HYP1,2-induced calcium influx (Fig 3A, left panel). We further examined whether the receptor mutants are affected in CEP14HYP1,2-induced MAPK activation. We found that cepr2 single mutants did not respond to CEP14HYP1,2-induced MAPK phosphorylation. By contrast, cepr1 and rlk7 mutants did not display reduced CEP14HYP1,2-induced MAPK activation (Figs 3B and S3D).

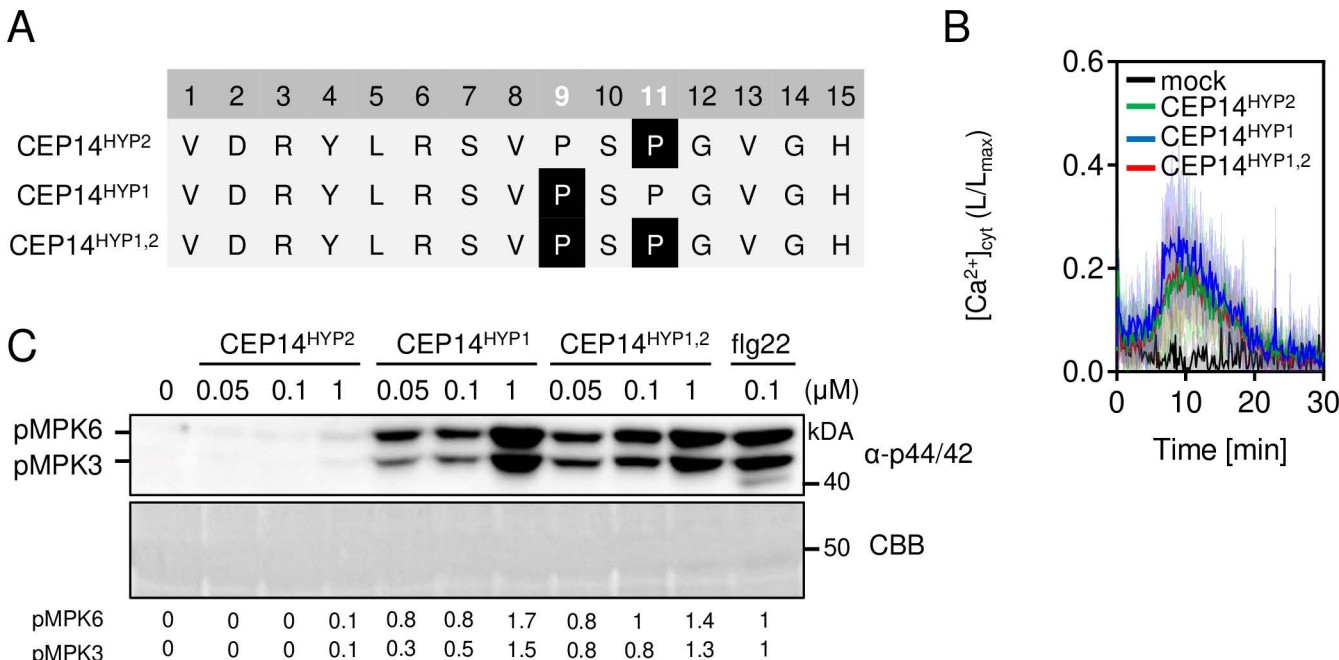

**Fig 2. CEP14 induces PTI-like responses in a hydroxyproline pattern-dependent manner.** A) Sequence alignment of CEP14 peptide variants used in this study. Proline residues are located at positions 9 and 11. Highlighted P indicates hydroxyproline residue. B) Kinetics of $[Ca^{2+}]_{cyt}$ in Col-0AEQ seedlings upon 1 µM CEP14 variants (n = 12) or mock treatment (n = 6) with mean ± SD. C) MAPK activation in Col-0 10 min upon CEP14 variants or flg22 treatment at indicated concentrations. Western blots were probed with α-p44/42. CBB = Coomassie brilliant blue. Band intensities of pMPK6 and pMPK3 were quantified and normalized to the Rubisco band (CBB stain) for each lane relative to flg22 treatment.

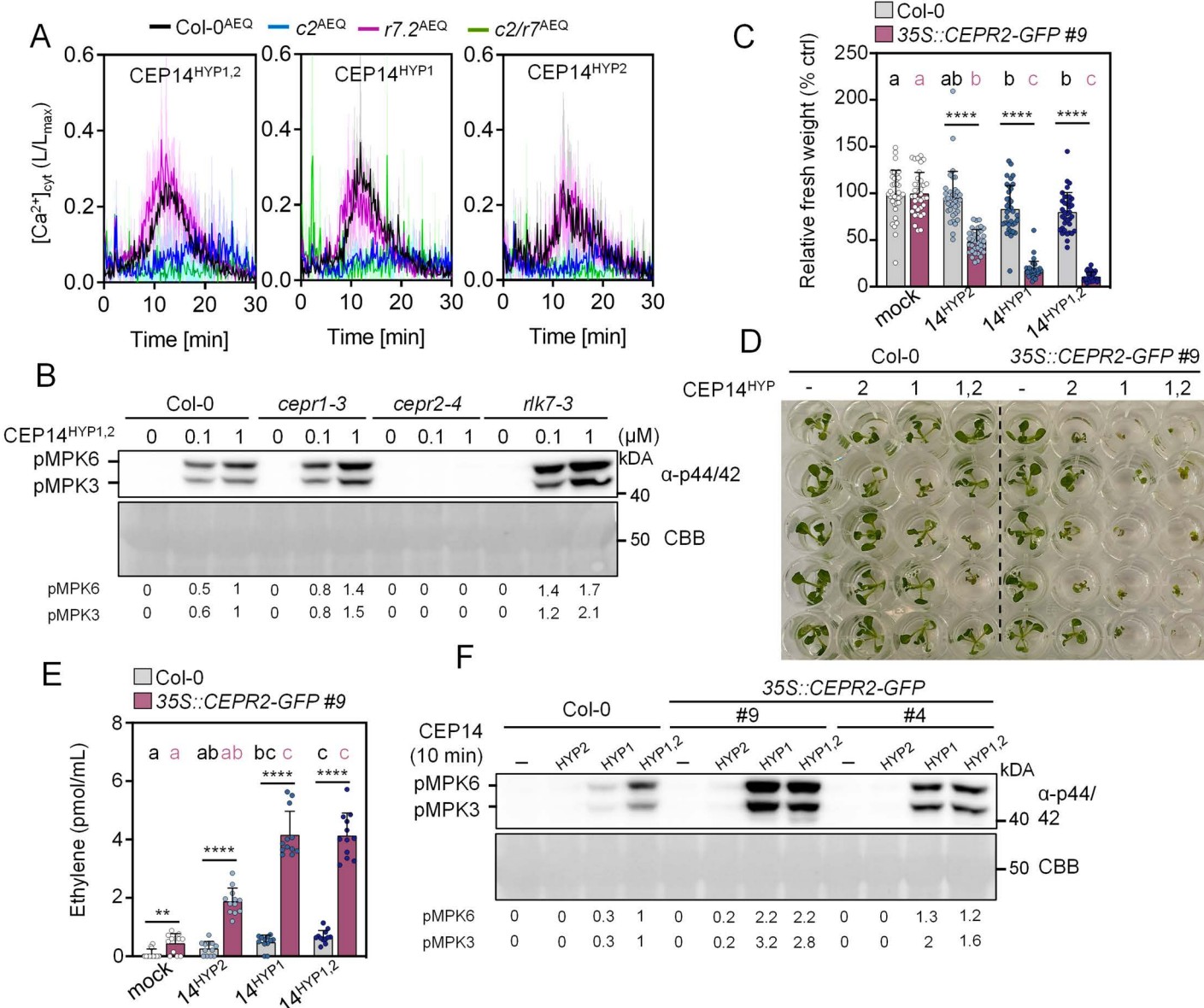

**Fig 3. CEP14 is primarily perceived by CEPR2.** A) $[Ca^{2+}]_{cyt}$ kinetics in seedlings upon CEP14 variant treatments (1 μM) in the indicated genotypes; n = 4 ± SD. **B)** MAPK activation was measured in the indicated genotypes 10 minutes after treatment with CEP14[HYP1,2] (0.1 and 1 μM). Western blots were probed with α-p44/42. CBB = Coomassie brilliant blue. Band intensities of pMPK6 and pMPK3 were quantified and normalized to the Rubisco band (CBB stain) for each lane relative to Col-0 CEP14[Hyp1,2] 1 μM treatment. **C)** Relative fresh weight of five-day-old seedlings treated with CEP14 variants (1 μM) for seven days; n = 36 pooled from three independent experiments, with mean ± SD. Black letters represent statistical comparisons between treatments in Col-0 (Kruskal-Wallis test, Dunn's post-hoc test, a-b, p < 0.01), while coral letters indicate comparisons between treatments in *35S::CEPR2-GFP* #9 (Kruskal-Wallis test, Dunn's post-hoc test, a-b, p = 0.0015, a-c, p < 0.0001, b-c, p ≤ 0.0001). Statistical comparisons between treatments across both genotypes were performed using the Mann-Whitney test (**** p < 0.0001). D) Representative image of C. E) Ethylene accumulation in leaf discs of the indicated genotypes 3.5 h upon mock (ddH₂O) or CEP14 variants (1 μM) treatment; n = 12 pooled from three independent experiments, with mean ± SD. Black letters represent statistical comparisons between treatments in Col-0 (one-way ANOVA, Tukey post-hoc test, a-bc, p = 0.0004, a-c, p < 0.0001, ab-c, p = 0.0003), while coral letters represent comparisons between treatments in *35S::CEPR2-GFP* #9 (Kruskal-Wallis test, Dunn's post-hoc test, a-c, p < 0.0001, ab-c, p ≤ 0.01). Statistical comparisons between treatments across both genotypes were performed using Welch's t-test (mock, CEP14[HYP2], **p = 00030, ****p < 0.0001) and Mann-Whitney test (CEP14[HYP1], ****p < 0.0001). **F)** MAPK activation 10 min upon CEP14 variants (100 nM) treatment in the indicated genotypes. Western blots were probed with α-p44/42. CBB = Coomassie brilliant blue. Band intensities of pMPK6 and pMPK3 were quantified and normalized to the Rubisco band (CBB stain) for each lane relative to Col-0 CEP14[Hyp1,2] treatment. Experiments in **A** and **C-F** were

repeated at least three times in independent biological repeats with similar results. The experiment in **B** was repeated once for 100 nM CEP14[HYP1,2] treatment (same mutants) and once with Col-0[AEQ], cepr1[AEQ], cepr2[AEQ] and rlk7[AEQ] with identical results ([S3C Fig]). Receptor dependency of 1 μM CEP14[HYP1,2] was only tested once (depicted).

CEP14[HYP2] did not induce seedling growth inhibition or ethylene accumulation, while CEP14[HYP1] and CEP14[HYP1,2] did trigger a mild but significant response in the wild-type plants ([Fig 3C]–[3E]). In line with the hypothesis that *CEPR2* is the primary class II CEP receptor, both CEP14-induced outputs were strongly enhanced in *35S::CEPR2-GFP* overexpression lines ([Fig 3C]–[3E]). The increased abundance of CEPR2 was particularly pronounced upon CEP14[HYP1] and CEP14[HYP1,2] treatment, which completely arrested seedling growth, caused strong ethylene accumulation, and enhanced MAPK activation ([Fig 3C]–[3F]). Collectively, our data suggest that CEPR2 is the primary receptor for CEP14, with a minor $Ca^{2+}$ influx-specific contribution of RLK7. We previously reported that *CEPR2* expression is strongest in stomatal guard cells, with weaker signals detected in the surrounding cell types [12]. However, we could not detect *CEP14* promoter activity in guard cells ([Fig 1F]). When further analysing *CEPR2* promoter activity around hydathodes in cotyledons, we noticed mVenus signals primarily in water pores and guard cells [12] ([S4 Fig]). This suggests that CEP14 may function in a predominantly non-cell autonomous manner.

### CEP13 and CEP15 trigger CEPR2-dependent immune outputs

Both CEP13 and CEP15 share a high degree of sequence similarity with CEP14, including the conserved Pro9 and Pro11 residues ([Fig 4A]). To assess whether other group II CEPs are able to induce immune-like responses, we obtained synthetic CEP13[HYP2], CEP13[HYP1,2], CEP15[HYP2] and CEP15[HYP1,2] peptides ([Fig 4A]). Both CEP13 variants and CEP15[HYP1,2] induced modest $Ca^{2+}$ influx, with double hydroxylated peptides displaying higher bioactivity ([Fig 4B] and [4C]). Additionally, CEP13[HYP1,2] induced strong MPK3/6 phosphorylation, similar to CEP14[HYP1,2], while CEP15[HYP1,2] elicited a much weaker response ([Fig 4D]). Importantly, MAPK activation in response to all three group II CEP[HYP1,2] peptides was non-detectable in the *cepr2–4* single mutant, suggesting a similar receptor dependency as observed for the CEP14 variants ([Fig 4D]). In line with this, the ethylene accumulation in response to different CEP13 and CEP15 peptide versions was significantly enhanced in the *CEPR2* overexpression lines, with the strongest effect observed for the double-hydroxylated peptides ([Fig 4E]). Interestingly, only CEP13[HYP1,2] significantly arrested wild-type seedling growth, and this effect was strongly amplified in the *35S::CEPR2-GFP* line ([Fig 4F]). In contrast, the CEP15[HYP1,2] treatment induced seedling growth inhibition only in the *CEPR2* overexpression line, whereas the CEP15[HYP2] failed to induce growth arrest in both wild-type and overexpression background ([Fig 4F]). Collectively, these findings suggest that, similar to CEP14, the related CEP13 and CEP15 peptides act as CEPR2 ligands to activate immune signalling, with their bioactivity being modulated by the proline hydroxylation pattern.

### Overexpression and loss-of-function studies demonstrate an important role of group II CEPs in antibacterial resistance

To investigate the role of group II CEPs in Arabidopsis immunity, we generated independent constitutive *CEP13, CEP14* or *CEP15* overexpression lines. We isolated three lines each of *35S::CEP14* and *35S::CEP15*, and two lines of *35S::CEP13* ([S5A]–[S5E Fig]). The overexpression lines showed an overall normal growth and shoot morphology, except for *35S::CEP14 #1*, *35S::CEP13 #4* and *35S::CEP13#7*, which were smaller compared to their respective wild-type controls (Col-0 and *pUBQ10::AEQ*) ([S5D Fig]). We also inspected primary root growth of the overexpression lines. Both *35S::CEP13* lines displayed reduced primary root elongation, while only one of three lines each for *35S::CEP14* and *35S::CEP15* displayed this phenotype ([Fig 5E]), suggesting a potential mild contribution to root growth regulation for CEP13, as previously anticipated [16]. All three independent *35S::CEP14* lines exhibited significantly enhanced resistance

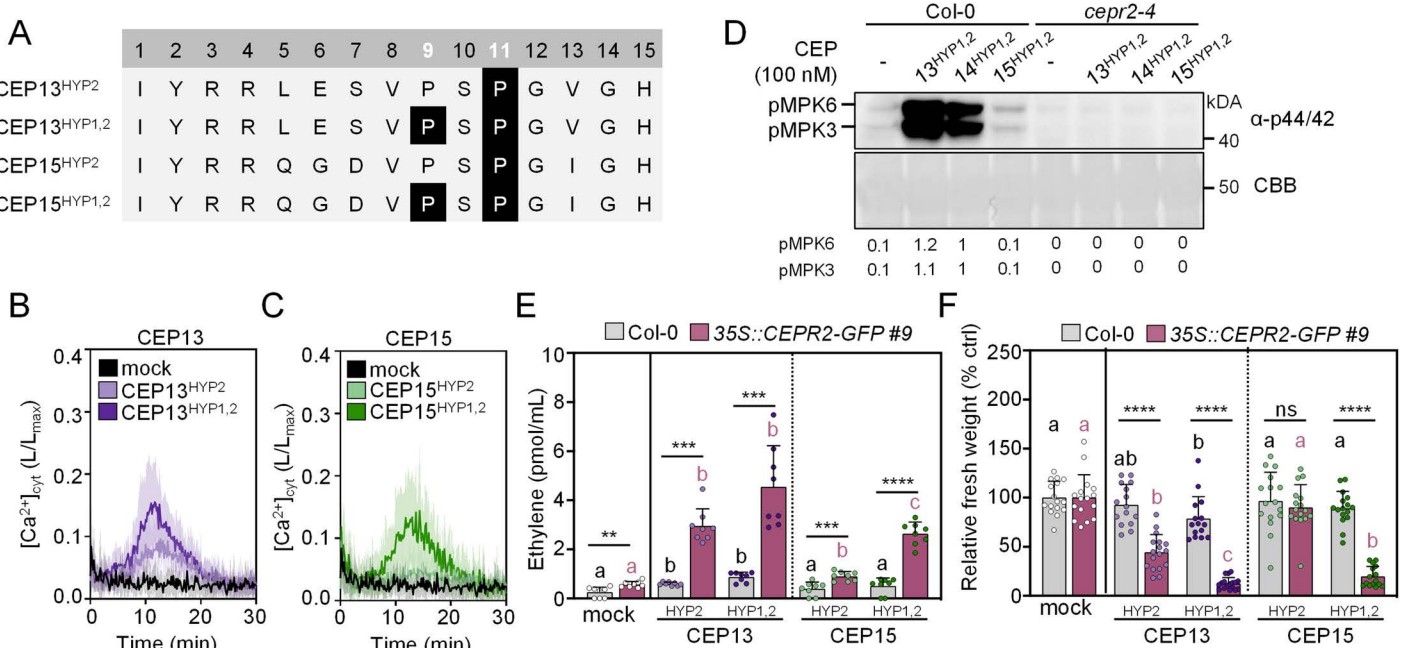

**Fig 4. CEP13 and CEP15 induce CEPR2-dependent PTI responses.** A) Sequence alignment of CEP13 and CEP15 peptide variants used in this study. Proline residues are located at positions 9 and 11. Highlighted P indicates hydroxyproline residue. **B)** and **C)** Kinetics of $[Ca^{2+}]_{cyt}$ in Col-0$^{AEQ}$ seedlings upon 1 µM CEP13 and CEP15 variants treatment, respectively; n = 12 ± SD. **D)** MAPK activation 10 min upon CEP13$^{HYP1,2}$, CEP14$^{HYP1,2}$, and CEP15$^{HYP1,2}$ (100 nM) treatment in the indicated genotypes. Western blots were probed with α-p44/42. CBB = Coomassie brilliant blue. Band intensities of pMPK6 and pMPK3 were quantified and normalized to the Rubisco band (CBB stain) for each lane relative to Col-0 CEP14$^{Hyp1,2}$ treatment. **E)** Ethylene accumulation in leaf discs of the indicated genotypes 3.5 h upon mock (ddH$_2$O) or CEP variants (1 µM) treatment; n = 8 pooled from two independent experiments ± SD. The dotted line indicates a separate statistical comparison for CEP13 and CEP15 peptide variants with black letters indicating statistical comparisons in Col-0 (Kruskal-Wallis, Dunn's post-hoc, a-b p < 0.05), while coral letters denote comparisons in *35S::CEPR2-GFP* #9 for mock-CEP13 (Kruskal-Wallis, Dunn's post-hoc, a-b p < 0.05) and mock-CEP15 (Brown-Forsythe and Welch ANOVA, Dunnett's T3 post-hoc, a-b p = 0.0042, a-c, b-c p < 0.0001). Comparisons between treatments across genotypes used two-tailed Student's t-test (mock, ** p = 0.0023; CEP15$^{HYP2}$, *** p = 0.0007; CEP15$^{HYP1,2}$, **** p < 0.0001), Mann-Whitney test (CEP13$^{HYP2}$, *** p = 0.0002), and Welch's t-test (CEP13$^{HYP1,2}$, *** p = 0.0004). **F)** Relative fresh weight of five-day-old seedlings treated with CEP13 and CEP15 variants (1 µM) for seven days; n = 15-17 pooled from two independent experiments, with mean ± SD. The dotted line indicates a separate statistical comparison for CEP13 and CEP15 peptide variants with black letters indicating statistical comparisons in Col-0 (one-way ANOVA, Tukey post-hoc test, A-B, p < 0.05), while coral letters denote comparisons in *35S::CEPR2-GFP* #9 for mock-CEP13 (Kruskal-Wallis, Dunn's post-hoc, a-b, b-c, p < 0.005, a-c p < 0.0001) and mock-CEP15 (Kruskal-Wallis, Dunn's post-hoc, a-b, p < 0.0001). Comparisons between treatments across genotypes used two-tailed Student's t-test (CEP13$^{HYP2}$ and CEP15$^{HYP1,2}$, **** p < 0.0001, CEP15$^{HYP2}$, ns = not significant, p = 0.5030), and Mann-Whitney test (CEP13$^{HYP1,2}$, **** p < 0.0001). All experiments were repeated three times in independent biological repeats with similar results, except for experiment **E**, which was performed twice with similar results.

against spray-inoculated hypovirulent *Pto*$^{COR-}$ lacking the effector molecule coronatine, which is commonly used to assess phenotypes associated with leaf surface immunity [23,24]. This suggests that CEP14 functions as a positive regulator of immunity against leaf-infecting *Pto* (Fig 5A). In contrast, both *35S::CEP13* lines did not display changes in resistance upon *Pto*$^{COR-}$ infection (Fig 5B). However, *35S::CEP15* lines showed increased resistance in an expression level-dependent manner (Figs 5B and S5C).

To confirm the role of group II CEPs in antibacterial resistance, we generated CRISPR-Cas9 knockout mutants in which all three family members were eliminated. We isolated two independent combinations of mutant alleles (*cep13/14/15* #1 and *cep13/14/15* #2, S5F Fig). Neither of the independent knockout mutants exhibited visible changes in shoot morphology (S5D Fig). Importantly, *cep13/14/15* mutant lines displayed impaired defence against *Pto*$^{COR-}$, with significantly increased bacterial titers in both independent lines (Fig 5C). Notably, Wang et al. (2024) reported that *cep14* single

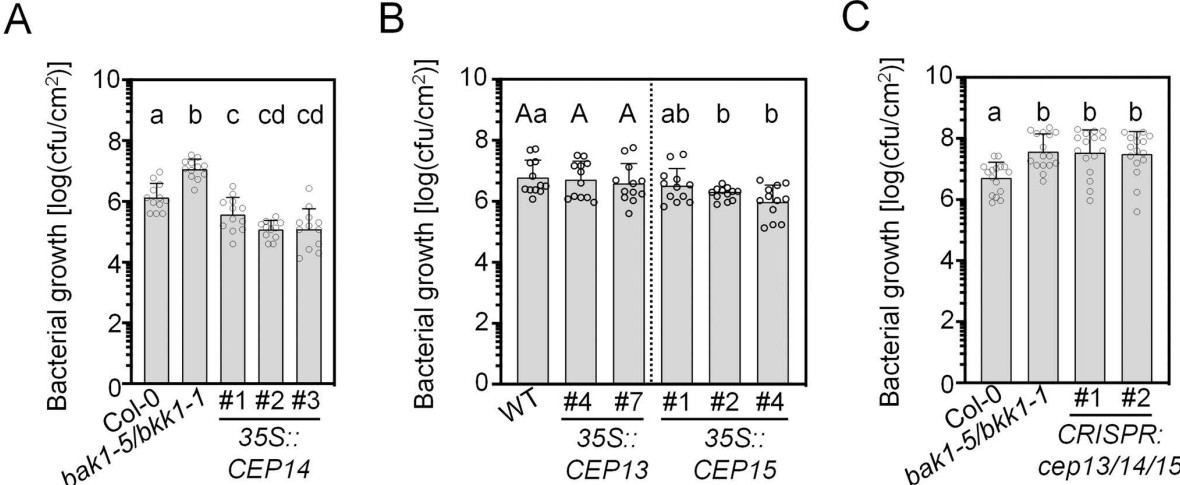

**Fig 5. Group II CEPs are important for resistance against *Pto*<sup>COR-</sup> infection.** A) Cfu of Pto<sup>COR-</sup> 3 dpi upon spray infection; n = 12 pooled from three independent experiments, with mean ± SD (one-way ANOVA, Tukey post-hoc test, a-b, a-cd, b-c, b-cd p < 0.0001, a-c p = 0.0149). B) Cfu of Pto<sup>COR-</sup> 3 dpi upon spray infection; n = 12 pooled from three independent experiments, with mean ± SD. Capital letters indicate statistical comparisons between Col-0 and *35S::CEP13* (one-way ANOVA, Tukey post-hoc test, not significant), while lowercase letters indicate comparisons between Col-0 and *35S::CEP15* (Brown-Forsythe and Welch ANOVA test, Dunnett's T3 post-hoc test, a-b p < 0.05). C) Cfu of Pto<sup>COR-</sup> 3 dpi upon spray infection; n = 16 pooled from four independent experiments, with mean ± SD (Kruskal-Wallis test, Dunn's post-hoc test, a-b, p < 0.01). All experiments were performed at least three times in independent biological replicates with similar results.

mutants did not display defects in resistance to *Pto* in infiltrated leaves, suggesting that CEP13/CEP15 may significantly contribute to disease resistance and/or stomatal immunity [18]. Collectively, this data, together with our group II *CEP* over-expression analysis, indicates that class II CEPs are important components of cell surface receptor-mediated immunity against bacterial infection in Arabidopsis with a more prominent role for CEP14 and CEP15.

### CEP14 peptides were not detected in apoplastic wash fluids, unlike CEP4

Given the observed differences in bioactivity among differentially hydroxylated synthetic class II CEP variants, we were interested to identify mature CEP14 peptides *in planta*. The CEP4 peptide was also included to determine which variant is the most abundant form detectable *in vivo* [12].

For this purpose, apoplastic wash fluids (AWFs) were extracted from both mature wild-type and *35S::CEP14* #1 lines with subsequent targeted mass spectrometry analysis. We included samples upon *Pto*<sup>COR-</sup> spray inoculation to test whether infection influences overall peptide accumulation and/or its proline hydroxylation pattern. In parallel, we aimed to identify the mature group I CEP4 variant that we previously characterized for its role in cell surface receptor-mediated immunity from wild-type and *35S::CEP4* #4 overexpression lines [12].

Synthetic reference peptides of the putative CEP14 variants (CEP14<sup>HYP1</sup>, CEP14<sup>HYP2</sup>, CEP14<sup>HYP1,2</sup>) were used for targeted analysis (Fig 6A). We similarly used synthetic CEP4 variants with 16 amino acid (CEP4<sup>16</sup>) and 15 amino acid (CEP4<sup>15</sup>) lengths that we previously characterized for their immune-inducing capabilities (Fig 6A) [12]. Targeted MS measurements of those peptides were recorded to establish reference parameters such as retention time and fragment ion intensities, which are required for accurate endogenous peptide variant identification (Figs 6C, 6F, S6A, S6D, and S6G).

Both CEP4 peptide variants displayed comparable MS intensities at similar concentrations, indicating an analogous detection sensitivity (Fig 6B). In contrast, the detection sensitivity for CEP14<sup>HYP2</sup> was nearly 100-fold lower than that for CEP4<sup>16</sup>. Among the CEP14 peptide variants, CEP14<sup>HYP1,2</sup> had the highest detection probability, although this was still lower

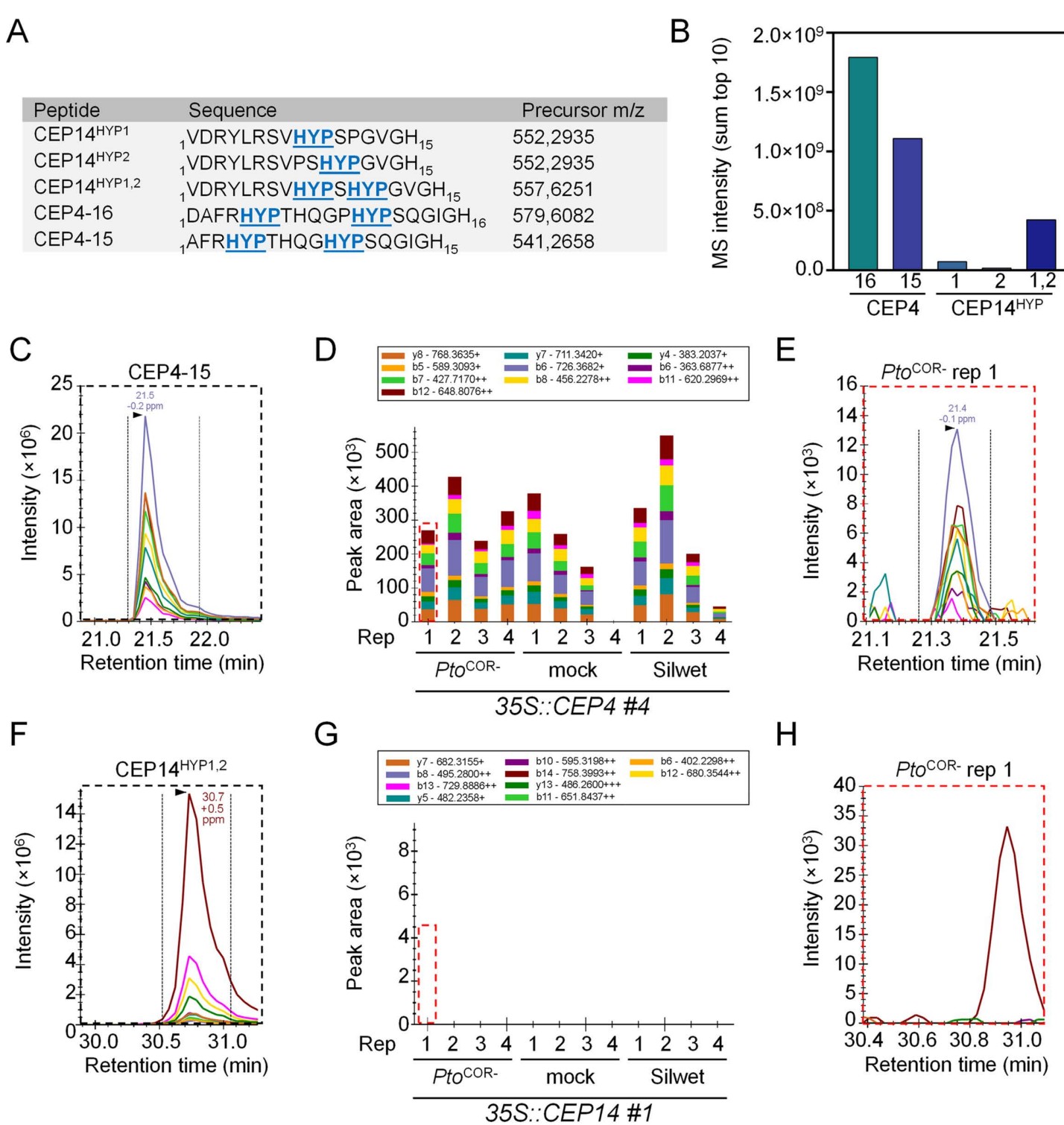

**Fig 6. CEP14HYP1,2 could not be detected in AWFs by targeted mass spectrometry.** A) List of synthetic CEP4 and CEP14 peptide variants, including amino acid sequences, Pro hydroxylation modifications (HYP), and precursor mass-to-charge (m/z) values used for targeted MS analysis. B) Bar graph showing detected peptide MS intensities (sum of the top 10 fragment ions) for synthetic CEP4 and CEP14 variants. All synthetic peptides were measured at a concentration of 450 fmol using Parallel Reaction Monitoring. C) Extracted ion chromatograms of synthetic CEP4[15] peptide. D) Quantification of CEP4[15] in AWFs from Pto[COR]-treated, mock-treated, and Silwet-treated 35S::CEP4 #4 plants (four biological replicates per condition). CEP4[15] was

endogenously detectable in 11 out of 12 tested samples, but no statistically significant regulation between treatments could be observed. **E)** Exemplary data of CEP4[15] detected in AWF from mock-treated *35S::CEP4* #4 rep 1 plants. **F)** Extracted ion chromatograms of synthetic CEP14[HYP1,2] peptide. **G)** Quantification of CEP14[HYP1,2] in AWFs from *Pto*[COR-]-treated, mock-treated, and Silwet-treated *35S::CEP14* #1 plants (four biological replicates per condition). CEP14[HYP1,2] was not endogenously detectable. **H)** Exemplary data of endogenous CEP14[HYP1,2] in AWF from *Pto*[COR-]-treated *35S::CEP14* #1 rep 1 plants. The dotted red squares in **D** and **G** highlight the chromatogram examples shown in **E** and **H**, respectively. The area under the curve of each coloured chromatogram reflects the MS intensity of a specific fragment ion. The sum of the 10 most intense fragment ions reflects the peptide MS intensity.

than CEP4[16] and CEP4[15] (Fig 6B). Nevertheless, the analysis of synthetic peptides indicated that all tested variants were detectable in plant samples if present at sufficiently high concentrations.

Despite the highest detection sensitivity, CEP4[16] was only detectable in two out of 12 analysed samples in *35S::CEP4* #4 samples (S6B and S6C Fig). By contrast, CEP4[15] was confidently identified in 11 out of 12 *35S::CEP4* #4 samples and absent in all AWFs isolated from wild-type plants with no apparent regulation of peptide concentrations following different treatments (Fig 6C–6E). These findings suggest that CEP4[15] is the predominant version of the CEP4 peptide found *in planta,* but accumulates at levels below the detection limit in wild-type plants. Consistent with this, we previously showed that CEP4[15] and CEP4[16] display identical bioactivity [12]. By contrast, none of the tested CEP14 peptide variants were detectable in the AWFs isolated from *35S::CEP14* #1 plants (Figs 6F–6H and S6D–S6I). Thus, the true nature of the CEP14 peptide *in planta* remains to be identified.

## Discussion

CEPs emerge as important regulators of multiple plant physiological processes, coordinating growth-related signals with biotic and abiotic stress responses [15]. Our work sheds new light on the complexity of CEP signalling in the regulation of plant immunity. We previously demonstrated that group I CEP peptides are important regulators of cell surface receptor-mediated immunity and are perceived by three distinct receptors. Canonical group I CEPs are perceived by CEPR1 and CEPR2, while the non-canonical member CEP4 additionally requires RLK7 for full-strength signal induction [12]. Notably, none of the previously analysed group I *CEP* genes showed strong transcriptional regulation during biotic stress. We now identified the group II CEP member *CEP14* to be induced upon elicitor treatment and *Pto* infection, a unique feature among all *CEP* genes.

We further reveal that CEP14 activates dose-dependent PTI outputs, including MAPK activation, $Ca^{2+}$ influx, seedling growth inhibition, and ethylene biosynthesis, similar to group I CEPs [12]. Interestingly, a recent study also described that pathogen-induced CEP14 activates PTI responses in Arabidopsis [18], supporting our findings. Beyond CEP14, our work revealed that other group II CEPs are also capable of inducing immune outputs, but in contrast to *CEP14,* their expression is not significantly modulated after elicitor treatment. Only the expression of *CEP15* is slightly repressed upon bacterial infection, raising the question whether *CEP15* transcript abundance is actively suppressed by *Pto* DC3000. Our data suggests that CEP14 is predominantly associated with group II-mediated immune regulation.

Both our study and the recent report [18] highlight that the bioactivity of group II CEPs is dependent on their proline hydroxylation pattern and that these peptides are perceived by CEPR2. This is unique among CEP family members since several canonical group I CEPs bind to CEPR1 and CEPR2 but primarily depend on CEPR1, while non-canonical CEP4 directly binds to the extracellular domain of CEPR2 and RLK7 and additionally depends on CEPR1 for signalling [12–15,25,26]. In line with the proposed role of CEPR2 as a receptor for apoplastic CEP14 perception, CEP14 carries a signal peptide for secretion and Co-IP experiments recently demonstrated interactions between CEPR2 and a full-length CEP14 peptide precursor [18]. However, the direct binding affinity of CEP14 to the extracellular domain of CEPR2, particularly in the context of proline hydroxylation or other posttranslational modifications, requires further investigation. The distinct receptor specificity among CEP ligands is likely sequence-driven. Notably, both non-canonical group I CEP4/CEP12

and group II CEPs (CEP13–15) share a conserved Pro9 residue, hydroxylation of which strongly promoted CEP14 bioactivity (Fig 2). Since *CEPR2-GFP* overexpression can enhance both CEP4 and CEP14 sensitivity (Fig 3) [12], Pro9 may be a key determinant for CEPR2 recognition. Based on peptide alignment, group II CEPs show a high degree of sequence similarity with PIPs. It is thus unexpected that they preferentially utilize CEPR2 to relay signals. It will be interesting to test in the future which distinct sequence features among CEPs and PIPs determine specificity to engage CEPR2, CEPR1 or RLK7. Our functional assays further suggest a minor, output-specific role for RLK7 in CEP14 signalling. CEP14[HYP2] and CEP14[HYP1,2]-induced $Ca^{2+}$ influx was strongly reduced in *cepr2* mutants and further reduced upon additional mutation of *RLK7*. However, MAPK activation was completely lost in *cepr2* single mutants, indicating that CEPR2 is essential for this signalling output. This raises the question whether RLK7 may fulfill a regulatory $Ca^{2+}$ influx-specific role rather than being directly involved in peptide binding. A similar phenomenon was described for the SERINE-RICH ENDOGENOUS PEPTIDEs receptor MALE DISCOVERER 1-INTERACTING RECEPTOR-LIKE KINASE 2, which contributes to PEP1 and flg22-induced ROS production but is dispensable for PEP1-mediated root growth inhibition [9,10].

Our targeted MS analysis failed to detect any CEP14 variants in apoplastic wash fluid samples under the tested conditions. This may be related to significant deviations of the *in vivo* peptide from our predicted CEP14 variant standards. We cannot exclude the presence of additional post-translational modifications or, alternatively, the lack of any hydroxylated proline residue *in vivo*. In *Medicago truncatula*, tri-arabinosylation on group I CEPs and different peptide fragment lengths have been previously demonstrated but these variants displayed reduced bioactivity. Moreover, non-hydroxylated CEP14 does not induce MAPK activation [18,20,27]. This prevents definitive conclusions about the *in planta* form of CEP14 and highlights the need for further investigation, such as untargeted and unbiased analysis of CEP14 peptide variants by mass spectrometry as well as improved purification and enrichment strategies for endogenous peptides. In most cases the identity of *in vivo* mature peptide variants is unknown, an important subject for future research. Functionally, neither CEP14-induced $Ca^{2+}$ influx nor MAPK activation were altered in *cepr1* mutants, suggesting that group II CEPs represent a unique CEPR1-independent clade within the CEP family. Consistently, CEP14[HYP2] retains vasculature binding in the absence of CEPR1 [28]. By contrast, the full-length CEP14 preprotein was shown to interact with CEPR1 in co-immunoprecipitation experiments, although this may reflect non-specific binding of CEPR1 to the precursor form rather than to the mature CEP14 peptide [18]. The identification of the bona fide CEP14 *in vivo* by more sensitive mass spectrometry approaches and disentangling receptor and ligand specificity among CEPs, PIPs and CEPR1/CEPR2/RLK7 will be an important future task.

Most analysed CEP ligands to date exhibit distinct tissue-specific expression patterns. Many CEP family members show activity in lateral root emergence zones, aligning with their known function in regulating root system architecture [14,29]. Above-ground CEP promoter activity varies across tissues, with signals detected in flowers, siliques, inflorescence stems, leaves, cotyledons, shoot apical meristems, hydathodes, and vasculature [12,14,29]. Our data now extend this knowledge by defining the expression domains of *CEP14* in the leaves. High-resolution analyses of a *pCEP14::NLS-3xmVenus* line revealed promoter activity in the epidermis, mesophyll, and the hydathode region in cotyledons, resembling the recently reported *pCEP4* activity pattern [12]. Promoter signal was further enhanced after flg22 treatment, consistent with our RT-qPCR-based data and in a recently published *pCEP14::GUS* reporter line [18], suggesting that the promoter construct accurately reflects endogenous *CEP14* regulation. Notably, *CEP14* promoter activity was absent in stomatal guard cells, both under control conditions and following flg22 treatment. In contrast, the CEP14 receptor *CEPR2* showed the highest expression in stomatal guard cells, with only weak signals in surrounding cells [12] (S4 Fig). In our *promoter::NLS-mVenus*-based analysis, *CEPR2* displayed weak expression around hydathodes, with the fluorescent signal being largely restricted to water pores (S4 Fig). This spatial separation between *CEP14* and *CEPR2* promoter activities raises the possibility that CEP14 may be mobile between cell types and/or tissues and function as a paracrine signalling molecule. Supporting this hypothesis, group I CEPs have been shown to act as long-distance signals coordinating adaptive growth responses during N starvation [14]. Similarly, CLE25 acts as a root-to-shoot signal that mediates abscisic acid (ABA)-induced stomatal closure during

dehydration stress via BAM receptors [30], further highlighting the potential of small signalling peptides to facilitate paracrine or systemic communication in plants.

CEP14-CEPR2 signalling was recently shown to be involved in systemic acquired resistance. While *cepr2* mutants displayed impaired SAR following *Pto* infection, the effect in *cep14* mutants was milder and limited to the hypovirulent *Pto*[COR-] strain [18]. This suggests functional redundancy at the ligand level and may result from a contribution of CEP13 and CEP15 to SAR, which we identified as CEPR2-dependent activators of immune responses. The role of CEP signalling in the regulation of SAR is further supported by our previous findings, where both a higher-order group I *CEP* mutant and the *cepr1/2* double mutant were compromised in *Pto avrRPM1*-triggered SAR [12]. Our work now extends the role of group II CEP signalling in plant immunity. The *cep13/14/15* triple mutants showed enhanced susceptibility to *Pto*[COR-] upon spray infection, indicating defects in cell surface receptor-mediated immunity. However, *cepr2* single mutants did not exhibit altered *Pto*[COR-] colonization, raising the question whether additional receptors may contribute to group II CEP perception during infection [12]. The mechanism by which CEP14 promotes Arabidopsis defence remains unknown. Stomata, crucial for gas exchange and water regulation, are targeted by leaf-invading microbes as entry points, driving the evolution of complex and dynamic mechanisms in both plants and pathogens to control their aperture. Upon detecting initial infection, plants respond to elicitors by rapidly closing stomata to restrict pathogen entry, a defence mechanism known as stomatal immunity [31]. Several phytocytokine pathways, including PIP1-RLK7 and PEP1-PEP RECEPTOR 1 (PEPR1) and PEPR2, have been shown to promote stomatal closure upon bacterial invasion [32–34]. It will be interesting to reveal whether CEP14 and/or CEP13/CEP15 can similarly promote stomatal closure to prevent pathogen entry. Importantly, stomatal aperture must be dynamically regulated. While prolonged closure restricts pathogen entry, it also creates a water-saturated apoplast that favours bacterial multiplication. Additionally, reduced transpiration caused by closed stomata limits photosynthetic carbon assimilation. To mitigate these effects, plants reopen stomata to disrupt the pathogen-favourable aqueous environment and support apoplastic immunity. This type of dynamic regulation was recently demonstrated for HAESA-LIKE 3/NUT-CTNIP/SCREW signalling pathway, which counteracts stomata closure during MAMP and ABA perception, optimizing plant fitness under stress [8,11]. Interestingly, CEPR2 directly phosphorylates ABA receptors and transporters, thereby suppressing ABA signalling and its translocation within the plant [35–37]. Although the role for CEPR2 in stomatal regulation has not yet been established, it is tempting to speculate that CEP14 perception may influence CEPR2 capability to modulate ABA-induced stomatal closure as part of the immune response. Future studies should address the mechanistic basis of CEP14-mediated control of cell surface receptor-mediated immunity and its interplay with SAR regulation.

CEP14 promoter activity was particularly strong at hydathodes, which can also serve as entry points for specific adapted pathogens [38]. These guttation organs secrete water through specialized pores in the epidermis or at leaf margins. Unlike stomata, hydathodes cannot fully close, but their aperture and exudation activities are regulated during infections and contribute significantly to hydathode immunity [38–40]. Similarly, *CEP4* promoter activity was also pronounced around the hydathode region [12]. It will be interesting to explore whether CEP4/CEP14 signalling plays a regulatory role during hydathode immunity, a largely understudied subject within the field of plant defence.

In summary, our work reveals a new role of CEP14 and related class II CEPs in the regulation of cell surface receptor-mediated immunity. A challenge for the future will be to disentangle the specific and overlapping functions of group I CEPs, group II CEPs and PIPs, particularly in light of their interactions with their distinct repertoire of endogenous receptors. Understanding how these peptide-receptor modules contribute to immune signalling and other physiological processes will be essential to fully grasp their roles in plant development and stress adaptation.

## Materials and methods

### Molecular cloning

To generate *CEP14* overexpression lines, the coding sequence of *CEP14* (AT1G29290) was synthesized (Twist Bioscience, USA) with attB attachment sites for Gateway cloning into pDONRZeo (Invitrogen, USA) and recombination with pB7WG2 (VIB, Ghent). To generate *CEP13* and *CEP15* overexpression lines, the coding sequence of *CEP13*

(AT1G16950) and *CEP15* (AT2G40530) were amplified with GoldenGate compatible primers containing BsaI overhangs and then cloned into a pUC18-derived vector, similar to previously described [41]. The respective coding sequences were then fused to the CaMV 35S promoter and terminator, and the resulting expression cassettes were assembled into a GoldenGate-adapted binary vector based on pCB302, together with a FastRed selection cassette allowing screening of transgenic seeds based on red fluorescence [42].

To generate CRISPR-Cas9 mutants, suitable target sites (two per gene) were selected using the online tool chopchop (https://chopchop.cbu.uib.no/). Gene-specific guide RNA (gRNA) constructs containing two target sites per gene (S1 Table) were synthesized (Twist Bioscience, USA) and cloned into a GoldenGate-adapted pUC18-derived vector. Individual gRNA constructs were stacked and subsequently assembled with FastRed-pRPS5::Cas9, into pICSL4723 for *in planta* expression [42]. The target sites used for generating CEPR2 and CEPR1 mutants were previously reported [12].

To generate the *pCEP14::NLS-3xmVenus* reporter construct, a 1026 bp promoter fragment upstream of the CEP14 start codon was amplified from genomic DNA with BsaI overhangs. The promoter was assembled with the sequence coding for nuclear localization signal of SV40 large T antigen followed by 3 sequential mVenus YFP fluorophores and a FastRed cassette for transgenic seed screening into a GoldenGate-modified pCB302 vector for plant expression, similar to previously described [12]. All final plant expression constructs were introduced into *Agrobacterium tumefaciens* strain GV3101 before Arabidopsis transformation via the floral dip. All primers used for cloning are listed in S2 Table.

## Plant material and growth conditions

Arabidopsis Col-0, Col-0$^{AEQ}$ (*p35S::AEQ*) and Col-0$^{U-AEQ}$ (*pUBQ10::AEQ*) were used as wild types backgrounds for experiments, as well as for the generation of transgenic lines and/or CRISPR mutants [43,44]. The *cepr1–3* (GK-467C01), *rlk7–3* (SALK_120595), and *cepr2–4* (GK-695D11) mutants were previously described and genotyped using gene-specific primers, as reported in earlier studies [7,12,25]. To isolate homozygous CRISPR mutants, T1 seeds carrying the pICSL4723 construct were selected based on red fluorescence and grown on soil. Genotyping was performed with gene-specific primers and Sanger sequencing. Transgene-free mutants were identified in subsequent generations by the loss of seed fluorescence.

Transgenic *35S::CEP14* lines were isolated based on Basta resistance, while *35S::CEP13* and *35S::CEP15* lines were identified based on red fluorescent seeds.

Plants were dark vernalized for 2–3 days at 4°C prior to transfer to growth conditions. For soil-based assays, plants were cultivated in pots under controlled environmental growth conditions (20–21°C, 55% relative humidity, 8 h photoperiod). For seedling-based assays, seeds were surface-sterilized with chlorine gas and germinated on ½ Murashige and Skoog (MS) media supplemented with vitamins (Duchefa, Netherlands), 1% sucrose, with or without 0.8% agarose. Seedlings were grown at 22°C under a 16 h photoperiod.

## Imaging and microscopy

Confocal laser-scanning microscopy was performed using a Leica TCS SP8 (Leica, Germany) microscope. To analyse promoter activity of *pCEP14::NLS-3xmVenus* lines, 10-day-old seedlings were transferred to a 24-well plate containing liquid 1/2 MS and treated with either ddH$_2$O (mock) or 1 μM flg22 and incubated for 16 hours. Similarly, untreated *pCEPR2::NLS-3xmVenus* 10-day-old seedlings were transferred to a 24-well plate containing liquid ½ MS 16 h prior to imaging. Propidium iodide staining (10 μg/mL) was applied immediately before microscopic analysis.

Images of NLS-3xmVenus signal in *pCEP14::NLS-3xmVenus* lines were acquired with argon laser excitation at 514 nm and a detection window of 516–558 nm. For *pCEPR2::NLS-3xmVenus* lines excitation was also performed at 514 nm, with a detection window of 516–565 nm. Propidium iodide was visualized using DPSS 561 laser emitting at 561 nm with a detection window of 610–630 nm. Laser intensity was adjusted according to the activity level of the promoters. Images were captured as maximal projections with 2 μm step size and processed with ImageJ V.1.54.

## Calcium influx assay

Apoaequorin-expressing liquid-grown eight-day-old seedlings were transferred individually to a 96-well plate containing 100 µL of 5 µM coelenterazine-h (PJK Biotech, Germany) and incubated in the dark overnight. Luminescence was measured using a plate reader (Luminoskan Ascent 2.1 or Varioskan LUX, Thermo Fisher Scientific, USA). Background luminescence was recorded by scanning each well 12 times at 10 s intervals before adding a 25 µl elicitor solution to the indicated final concentration. Luminescence was recorded for 30 min at the same interval. The remaining aequorin was discharged using a solution containing 2 M $CaCl_2$ and 20% ethanol. The values for cytosolic $Ca^{2+}$ concentrations ($[Ca^{2+}]_{cyt}$) were calculated as luminescence counts per second relative to total luminescence counts remaining ($L/L_{max}$).

## Ethylene measurement

Leaf discs (4 mm in diameter) from four-to-five-week-old soil-grown Arabidopsis were incubated post-harvest overnight in $ddH_2O$. Three leaf discs per sample were transferred to individual glass vials containing 500 µl $ddH_2O$ before adding $ddH_2O$ (mock) or peptides to the indicated final concentration. Glass vials were capped with rubber lids and gently agitated for 3.5 h. One mL of vial headspace was extracted with a syringe and injected into a Varian 3300 gas chromatograph (Varian, USA) to measure ethylene accumulation.

## MAPK activation and western blot analysis

Five-day-old Arabidopsis seedlings growing on ½ MS agar plates, were transferred into a 24-well plate containing liquid medium for seven days. 24 h before treatment, seedlings were equilibrated in fresh ½ MS medium. Six seedlings per sample were harvested, frozen in liquid nitrogen and homogenized using a tissue lyser (Qiagen, Germany). Proteins were extracted using a 50 mM Tris-HCl (pH 7.5) buffer containing 50 mM NaCl, 10% glycerol, 5 mM DTT, 1% protease inhibitor cocktail, 1 mM phenylmethylsulfonyl fluoride (PMSF), 1% IGEPAL, 10 mM EGTA, 2 mM NaF, 2 mM $Na_3VO_4$, 2 mM $Na_2MoO_4$, 15 mM ß-Glycerophosphate and 15 mM p-nitrophenylphosphate before analysis by SDS-PAGE and western blot. Phosphorylated MAPKs were detected using α-p44/42 primary antibodies (Cell Signaling, USA) and secondary mouse α-rabbit IgG-HRP antibodies (Santa Cruz Biotechnology, sc-2357). Band intensities of MPK6 and MPK3 were quantified using ImageJ and normalized to the Rubisco band (CBB stain) for each lane. Normalized values were expressed relative to the calibrator sample which was set to 1 and is indicated in the respective figure legend.

## Seedling growth inhibition

Arabidopsis seedlings were grown for five days on ½ MS agar plates before transfer of single seedlings into individual wells of a 48-well plate and liquid medium with or without elicitors. After seven-day treatment, fresh weight of individual seedlings was measured.

## Pathogen growth assay

*Pseudomonas syringae* pv. *tomato* lacking the effector coronatine ($Pto^{COR-}$) was grown on King's B agar plates containing 50 µg/mL rifampicin and 50 µg/mL kanamycin at 28ºC. After two-to-three days bacteria were resuspended in $ddH_2O$ containing 0.04% Silwet L77 (Sigma Aldrich, USA). The bacterial suspension was adjusted to an $OD_{600} = 0.2$ ($2 \times 10^8$ cfu/mL) before thoroughly spray-inoculating five-week-old Arabidopsis. Bacterial counts were determined three days after infection by re-isolation and plating of colonies with subsequent colony counting. Bacterial growth was calculated as $cfu/cm^2$ of leaf area.

## Gene expression analysis

For seedling-based assays, 12-day-old liquid-grown seedlings were equilibrated in fresh medium for 24 h before treatment with the indicated peptides. For adult plants, four-to-five-week-old Arabidopsis leaves were syringe-infiltrated with

ddH$_2$O (mock), flg22 (1 μM) or *Pto* DC3000 (OD$_{600}$ = 0.001, 5x10$^5$ cfu/mL) and incubated for 24 h. All samples for RT-qPCR analysis were harvested at the indicated time points, frozen in liquid nitrogen and ground with a tissue lyser (Qiagen, Germany). Total RNA was isolated with TRIzol reagent (Roche, Switzerland) and purified using Direct-zol RNA Miniprep Plus kit (Zymo Research, Germany). 2 μg of the total RNA was digested with DNase I and reverse transcribed with oligo (dT)18 and Revert Aid reverse transcriptase. RT-qPCR experiments were performed using Takyon Low ROX SYBR MasterMix (Eurogentec, Belgium) with the AriaMx Real-Time PCR system (Agilent Technologies, USA). Expression levels of all tested genes were normalized to the house-keeping gene *Ubiquitin 5* (*UBQ5*). Sequences of all primers used for RT-qPCR analysis are found in S2 Table.

## Detection of CEP4 and CEP14 peptide variants from apoplastic wash fluids

Seven-week-old soil-grown Col-0, *35S::CEP4* #4, and *35S::CEP14* #1 Arabidopsis plants were sprayed with either *Pto*$^{COR-}$, a ddH$_2$O solution containing 0.04% Silwet or left untreated. Afterward, the plants were covered with lids and returned to short-day growth chambers. 24 h post-treatment, plant shoots were harvested in groups of three and submerged in 100 mL of ice-cold apoplastic wash fluid buffer (5 mM sodium acetate, 0.2 M calcium chloride, pH 4.3), freshly supplemented with protease inhibitor cocktail (Roche). The samples were vacuum infiltrated two times (2 min each), and the AWFs were collected by centrifugation at 800 x g for 20 minutes at 4°C. The eluates were subjected to chlorophenol extraction [45] with the modification of substituting methylmorpholine for N-ethylmorpholine. Peptidome enrichment was performed using PD-Miditrap G-10 size exclusion gravity columns (Cytiva, Amersham, UK). The samples were lyophilized and resuspended in 500 μL of ddH$_2$O for subsequent analysis.

The CEP4 and CEP14 peptides were analysed directly from AWFs by targeted mass spectrometry using Parallel Reaction Monitoring (PRM). Prior to the analysis, synthetic CEP4 and CEP14 reference peptides were used to establish and optimize PRM assays using a final concentration of 450 fmol (on column) per synthetic peptide. PRM data acquisition was performed with a 50-min linear gradient on a Dionex Ultimate 3000 RSLCnano system coupled to a Fusion Lumos mass spectrometer (Thermo Fisher Scientific). MS1 spectra (360–1300 m/z) were recorded at a resolution of 120,000 using an automatic gain control (AGC) target value of $4 \times 10^5$ and a maximum injection time (MaxIT) of 100 msec. Targeted MS2 spectra were acquired in the Orbitrap at 60,000 resolution using higher-energy collisional dissociation (HCD) with a 30% normalized collision energy (NCE), an AGC target value of $2 \times 10^5$, a MaxIT of 118 msec and a Quadrupole isolation window of 1.3 m/z. Within a single PRM run, maximally 20 peptide precursors were targeted with a retention time scheduling of 15 min. PRM data were analysed using the Skyline software (version Skyline-daily 64-bit 24.1.1449 [46,47]. From the synthetic peptide measurements, the 10 most intense transitions (precursor-fragment ion pairs) per peptide precursor, as well as retention time, were determined. Endogenous CEP4 and CEP14 peptide intensities in AWFs were manually evaluated and peptide intensities were calculated by summing of all transition intensities. If necessary, integration boundaries were manually adjusted and strongly interfered transitions were removed.

## Synthetic peptides

The flg22 peptide was kindly provided by Dr. Justin Lee (IPB Halle). Other peptides were synthesized by Pepmic (China) with at least 90% purity and dissolved in ddH$_2$O. Sequences of all peptides used in this study are displayed in S3 Table.

## Statistics and reproducibility

Statistical analysis was conducted using GraphPad Prism (10.2.3, 347). Data were evaluated for Gaussian distribution using the Shapiro-Wilk normality test and visual assessment of QQ plots. Homogeneity of variance was assessed using the F test for paired analyses or the Brown-Forsythe test for multiple comparisons. If both normality and homogeneity of variance were met, parametric tests were applied. For comparisons between two groups, a two-tailed t-test was used. If

data followed a Gaussian distribution but variances between groups were unequal, an unpaired t-test with Welch's correction was performed. For non-Gaussian data, the Mann-Whitney U test was applied.

For multiple comparisons of normally distributed data with equal variances, one-way ANOVA was followed by Tukey's post-hoc test for pairwise comparisons or Dunnett's post-hoc test for comparisons against a single control group. When variances were unequal, Welch's ANOVA with Dunnett's T3 multiple comparison test was employed. If data did not follow a Gaussian distribution, the Kruskal-Wallis test was used, followed by Dunn's multiple comparison test. Details on sample size, p-values and/or ranges, the number of biological replicates, and statistical methods employed are provided in the respective figure legends.

## Supporting information

**S1 Fig. Group II CEP transcriptional regulation during biotic stress. A)** Tissue-specific expression potential of class II *CEP* family members. **B)** *CEP* expression after flg22 treatment and *Pto*/*Pseudomonas syringae* pv. *maculicola (Psm)* infection. **C)** Expression heat map of class II *CEPs* in wild-type seedlings following mock, flg22, elf18, PEP1, nlp20, OGs and CO8, treatment for the indicated time. $Log_2$ (fold change) relative to time-point 0 h for each treatment. The data was obtained from Bjornson et al., 2021 [14]. **D)** Comparison of CEP and PIP putative mature peptide sequences aligned with the MAFFT tool. Conserved sequence features are highlighted to illustrate similarities and distinctions among the families. The C-terminal GxGH motif and Ser10 are shaded in dark grey, with the variable "x" position among CEPs and PIPs shown in light grey. Specific residues are color-coded for clarity: all proline residues are highlighted in yellow, CEP4/CEP12-specific residues in turquoise, class II CEP-specific residues in light blue, and class I-specific residues are shown in purple. Residues shared between certain class II CEPs and PIPs are highlighted in dark blue, and residues shared between class I CEPs and PIP/PIPLs are highlighted in light green. The proline at position 9, which is present in CEP4, CEP12, CEP13, CEP14, CEP15, and PIP/PIPLs but absent in canonical class I CEPs, is highlighted in bold and framed. Data in **A** and **B** were obtained using Genevestigator software and are based on the AT_mRNASeq_ARABI_GL-1 data set.
(TIF)

**S2 Fig. CEP14$^{HYP2}$ is a weak inducer of PTI-like responses: A) MAPK activation in Col-0 10 min upon CEP14$^{HYP2}$ treatment at the indicated concentrations.** Western blots were probed with α-p44/42. CBB = Coomassie brilliant blue. **B)** Ethylene accumulation in Col-0 leaf discs 3.5 h upon mock (ddH$_2$O) or CEP14$^{HYP2}$ (1 μM) treatment; n = 17 pooled from four independent experiments, with mean ± SD (Mann-Whitney test, p = 0.2012). **C)** Relative fresh weight of five-day-old seedlings treated with CEP14$^{HYP2}$ (1 μM) for seven days; n = 48 pooled from four independent experiments, with mean ± SD (two-tailed Student's t-test, p = 0.4974). All experiments were performed at least three times in independent biological replicates with similar results.
(TIF)

**S3 Fig. Characterization of cepr1$^{AEQ}$ and cepr2$^{AEQ}$.** Schematic diagram of the *CEPR1* **(A)**, *CEPR2* **(B)** and *RLK7* **(C)** genomic sequence, structure, and the CRISPR-Cas9-mediated mutation pattern detected by DNA sequencing. The locus number and the length of the coding sequence (CDS) are indicated above the scheme for each gene. The CRISPR cepr1$^{AEQ}$ single mutant (with CRISPR allele *cepr1.2$^{AEQ}$*) was generated in Col-0$^{AEQ}$ background with a CRISPR *cepr1* construct. The CRISPR *cepr2.2$^{AEQ}$* mutant was generated with a construct targeting both *CEPR1* and *CEPR2* [12]. The cepr2 rlk7$^{AEQ}$ line was generated in the *cepr2.2$^{AEQ}$* background with a previously described *rlk7* construct [12]. The specific location and type of mutations for each gene are indicated in the schematics describing the mutants. The two CRISPR target sites are indicated in purple, exons are indicated in grey, and introns are shown as black lines; scale bar = 200 bp. **D)** MAPK activation in indicated genotypes 10 min upon CEP14$^{HYP1,2}$ treatment (100 nM). Western blots were probed with

α-p44/42. CBB = Coomassie brilliant blue. Band intensities of pMPK6 and pMPK3 were quantified and normalized to the Rubisco band (CBB stain) for each lane relative to Col-0[AEQ] CEP14[Hyp1,2] treatment.
(TIF)

**S4 Fig. CEPR2 promoter activity around hydathodes.** Representative images of NLS-3xmVenus signal in *pCEPR2::NLS-3xmVenus* lines around hydathodes and the epidermis. The maximum projection of Z-stacks for mVenus is merged with the Z-stacked PI signal; scale bar = 50 µm. The experiment was performed three times in independent biological repeats with similar results.
(TIF)

**S5 Fig. Characterization of group II CEP overexpression lines and loss of function mutants. A)** *CEP14* transcript levels in three independent *CEP14* overexpression lines, shown as fold expression compared to Col-0. Housekeeping gene *UBQ5*; n = 4 with mean ± SD. **B)** *CEP13* transcript levels in two independent *CEP13* overexpression lines, shown as fold expression compared to wild-type (aequorin reporter line, *pUBQ10::AEQ*). Housekeeping gene UBQ5; n = 3 with mean ± SD. **C)** *CEP15* transcript levels in three independent *CEP15* overexpression lines, shown as fold induction compared to wild-type (aequorin reporter line, *pUBQ10::AEQ*). Housekeeping gene *UBQ5*; n = 3 with mean ± SD. **D)** Pictures of 5-week-old plants of the indicated genotypes grown on soil. Sale bar = 1 cm. **E)** Seedlings were grown on 1/2 MS phytagel plates under long-day conditions. Germinated seedlings were transferred to fresh plates and imaged after 5 days. The primary root length was measured using Fiji (ImageJ) with the NeuronJ plug-in. Data represent mean ± SD from n = 32–33 pooled from independent experiments. The dotted line indicates separate experiments and statistical comparisons (one-way ANOVA, Tukey post-hoc test for *CEP14* lines a-b p < 0.0001; for *CEP13* lines a-b p < 0.0001, a-c p < 0.0001, b-c, p = 0.0078; for *CEP15* lines WT vs #2 a-b, p = 0.0144, #1 vs #2 p = 0.0159). **F)** Characterization of two independent *cep13/14/15* mutants, CRISPR *cep13 cep14 cep15* #1 with alleles *cep13.1*, *cep14.1*, *cep15.1* and #2 mutant with alleles *cep13.2*, *cep14.2*, *cep15.2*. Schematic diagram of *CEP13*, *CEP14* and *CEP15* gene structure and the CRISPR-Cas9-mediated mutation pattern detected by DNA sequencing. The locus number and the length of the CDS are indicated above the scheme for each gene. The specific location and type of mutations for each gene are indicated in the schematics describing the mutants. The CEP domain is indicated in blue, and the two CRISPR target sites are indicated in purple. Scale bar = 100 bp.
(TIF)

**S6 Fig. CEP4[16] shows limited detectability *in planta*, and CEP14 peptide variants could not be detected by targeted mass spectrometry. A)** Extracted ion chromatograms of synthetic CEP4[16] peptide. **B)** Quantification of CEP4[16] in AWFs from *Pto*[COR-]-treated, mock-treated and Silwet-treated *35S::CEP4* #4 plants (four biological replicates per condition). CEP4[16] was endogenously detectable in only two out of 12 tested samples. **C)** Exemplary data of CEP4[16] peptide detected in AWF from *Pto*[COR-]-treated *35S::CEP4* #4 rep 1 plants. **D)** Extracted ion chromatograms of synthetic CEP14[HYP1] peptide. **E)** Quantification of CEP14[HYP1] in AWFs from *Pto*[COR-]-treated, mock-treated and Silwet-treated *35S::CEP14* #1 plants (four biological replicates per condition). CEP14[HYP1] was not endogenously detectable. **F)** Exemplary data of endogenous CEP14[HYP1] in AWFs from *Pto*[COR-]-treated *35S::CEP14* #1 rep 1 plants. **G)** Extracted ion chromatograms of synthetic CEP14[HYP2] peptide. **H)** Quantification of CEP14[HYP2] in AWFs from *Pto*[COR-]-treated, mock-treated and Silwet-treated *35S::CEP14* #1 plants (four biological replicates per condition). CEP14[HYP2] was not endogenously detectable. **I)** Exemplary data of endogenous CEP14[HYP2] in AWFs from *Pto*[COR-]-treated *35S::CEP14* #1 rep 1 plants. All synthetic peptides were measured at a concentration of 450 fmol using Parallel Reaction Monitoring. The dotted red squares in **B**, **E** and **H** highlight the chromatogram examples shown in **C**, **F** and **I**, respectively. The area under the curve of each coloured chromatogram reflects the MS intensity of a specific fragment ion. The sum of the 10 most intense fragment ions reflects the peptide MS intensity.
(TIF)

**S1 Table. CRISPR target sites used in this study.**
(XLSX)

**S2 Table. Primers used in this study.**
(XLSX)

**S3 Table. Peptides used in this study.**
(XLSX)

**S1 Data. Numeric data of all graphs depicted in Figs 1–5, S2, S3 and S5.**
(XLSX)

## Acknowledgments

We thank Stefanie Ranf for providing GoldenGate vectors for molecular cloning and Mark Youles (TSL Norwich) and Laurence Tomlinson (TSL Norwich) for providing the pICSL4723OD vector for CRISPR-Cas9 cloning. We thank Radim Sarlej and Karin Dengler-Wupperfeld for technical assistance.

## Author contributions

**Conceptualization:** Jakub Rzemieniewski, Martin Stegmann.

**Funding acquisition:** Christina Ludwig, Martin Stegmann.

**Investigation:** Jakub Rzemieniewski, Patricia Zecua-Ramirez, Sebastian Schade, Zeynep Camgöz, Genc Haljiti, Sukhmannpreet Kaur, Christina Ludwig.

**Methodology:** Jakub Rzemieniewski, Genc Haljiti, Christina Ludwig.

**Project administration:** Martin Stegmann.

**Supervision:** Jakub Rzemieniewski, Christina Ludwig, Ralph Hückelhoven, Martin Stegmann.

**Validation:** Patricia Zecua-Ramirez, Sebastian Schade.

**Writing – original draft:** Jakub Rzemieniewski, Martin Stegmann.

**Writing – review & editing:** Jakub Rzemieniewski, Patricia Zecua-Ramirez, Sebastian Schade, Genc Haljiti, Christina Ludwig, Ralph Hückelhoven.

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
