## [Decision Letter · Decision Letter 0]

8 May 2025

CEPR2 perceives group II CEPs to regulate cell surface immunity in Arabidopsis

PLOS Pathogens

Dear Dr. Martin Stegmann,

Thank you for submitting your manuscript to PLOS Pathogens. Your manuscript has undergone a thorough evaluation at the editorial level and by independent peer reviewers. The reviewers recognized the significance of the problem addressed in your work but raised several substantial concerns regarding the manuscript in its current form. While there are some questions about the novelty of your findings in light of the recent publication by Wang et al. in Plant Physiology (2025), we are open to considering your manuscript for publication, provided that all other reviewer comments are adequately addressed. Therefore, we invite you to submit a revised version of the manuscript that responds to the points raised during the review process.

Please submit your revised manuscript within 60 days Jul 07 2025 11:59PM. If you will need more time than this to complete your revisions, please reply to this message or contact the journal office at plospathogens@plos.org. Please include the following items when submitting your revised manuscript:

We look forward to receiving your revised manuscript.

Kind regards,

Huiquan Liu

Guest Editor

PLOS Pathogens

Bart Thomma

Section Editor

Editor-in-Chief

PLOS Pathogens

Michael Malim

Editor-in-Chief

PLOS Pathogens

orcid.org/0000-0002-7699-2064

**Journal Requirements:**

At this stage, the following Authors/Authors require contributions: Jakub Rzemieniewski, Patricia Zecua-Ramirez, Sebastian Schade, Zeynep Camgöz, Genc Haljiti, Sukhmannpreet Kaur, Christina Ludwig, Ralph Hückelhoven, and Martin Stegmann. Please ensure that the full contributions of each author are acknowledged in the "Add/Edit/Remove Authors" section of our submission form.

https://journals.plos.org/plospathogens/s/submission-guidelines#loc-parts-of-a-submission

4) We noticed that you used the phrase 'data not shown' in the manuscript. We do not allow these references, as the PLOS data access policy requires that all data be either published with the manuscript or made available in a publicly accessible database. Please amend the supplementary material to include the referenced data or remove the references.

5) Please upload all main figures as separate Figure files in .tif or .eps format. For more information about how to convert and format your figure files please see our guidelines: 

6) We notice that your supplementary Figures, and Tables are included in the manuscript file. Please remove them and upload them with the file type 'Supporting Information'. Please ensure that each Supporting Information file has a legend listed in the manuscript after the references list.

7) Some material included in your submission may be copyrighted. According to PLOSu2019s copyright policy, authors who use figures or other material (e.g., graphics, clipart, maps) from another author or copyright holder must demonstrate or obtain permission to publish this material under the Creative Commons Attribution 4.0 International (CC BY 4.0) License used by PLOS journals. Please closely review the details of PLOSu2019s copyright requirements here: PLOS Licenses and Copyright. If you need to request permissions from a copyright holder, you may use PLOS's Copyright Content Permission form.

Potential Copyright Issues:

- Please confirm (a) that you are the photographer of Figures 3-E and S5-D, or (b) provide written permission from the photographer to publish the photo(s) under our CC BY 4.0 license.

8) Please amend your detailed Financial Disclosure statement. This is published with the article. It must therefore be completed in full sentences and contain the exact wording you wish to be published. Please ensure that the funders and grant numbers match between the Financial Disclosure field and the Funding Information tab in your submission form. Note that the funders must be provided in the same order in both places as well.

**Reviewers' Comments:**

Reviewer's Responses to Questions

**Part I - Summary**

Reviewer #1: In this study, the authors investigated the role of group II CEPs and their receptor CEPR2 to regulate cell surface immunity in Arabidopsis. Overall, this is a well presented body of work with clear evidence to support the conclusion. The findings are interesting and provide valuable information for researchers working on plant-pathogen interaction. However, some issues should be addressed before it accepted for publication on PLoS Pathogens.

Reviewer #2: This manuscript by Rzemieniewski et al. demonstrated the group II CEP peptides CEP14, CEP13, and CEP15 act as phytocytokines to trigger hallmark immune responses and contribute to plant resistance against phytobacteria. The perception of group II CEPs is mainly dependent on CEPR2. I found the main conclusion of this MS is relatively similar with the recently published work at Plant Physiology (Wang et al, Plant Physiology, 2025). Although this study extended the analysis to show CEP13 and CEP15 are also able to induce immunity, their functional peptides are relatively conserved and exhibit the similar mode of action with CEP14. In addition, RLK7 is shown to participate in the CEP14/CEPR2-mediated Ca2+ influx, how it is implicated in the receptor complex and specific for Ca2+ response are not determined.

Reviewer #3: In this manuscript, Rzemieniewski et al., present research on the role of group II CEPs in plant immunity, specifically focusing on CEP14 and its receptor CEPR2 in Arabidopsis. The study provides evidence that group II CEPs are involved in PTI-like responses, contribute to antibacterial resistance, and that CEPR2 is required for the CEP14 recognition. The results are presented in a concise manner, and they support the primary conclusions. Meanwhile, I have several comments that should be addressed in the revised manuscript. The following points outline these comments

**Part II – Major Issues: Key Experiments Required for Acceptance**

Reviewer #1: 1. Does CEPs, such as CEP14 specifically induced by flag22 and Pto? What about fungal pathogens or fungal PAMPs?

2. How CEPR2 perceives CEP13, CEP14 and CEP15? Via interaction or other ways? Please provide evidence.

3. How different groups of CEPs perceive different receptors?

4. Can you detect hydroxylated CEP14HYP2 in plant during pathogen infection? Or it can only be synthesized in vitro?

5. Figure 2B-C, likely missed a CEP14 control

6. Figure 5, please display the inoculated plants, I want to see the phenotype.

7. The CEPs-CEPR recognition mode is conserved in plant or not?

Reviewer #2: (No Response)

Reviewer #3: Major comments:

1. Lack of Direct Evidence for CEP14-CEPR2 Interaction: While the genetic and functional data suggest that CEPR2 is a key receptor for CEP14, there is no direct evidence of their physical interaction.

2. Line 223 states that CEP14 functions as a positive regulator of immunity against leaf-infecting pathogens, based on Pto COR- infection data. This is a significant claim that requires broader support. Given that Pto COR- is a lab-modified pathogen, it would be important to assess the effect of CEP14 overexpression against wild-type Pto, Psm, and potentially fungal pathogens to validate this claim more comprehensively.

3. The authors conclude that CEP14 plays a "particularly prominent role" in cell surface receptor-mediated immunity (lines 237-238). However, the data supporting this conclusion is not sufficiently robust. The PtoCOR- infection assays lack crucial controls. Specifically, the manuscript needs to include the analysis of cep14 single mutants alongside the cep13 and cep15 single mutants. The claim about CEP14's prominent role cannot be fully supported without a direct, side-by-side comparison of these single mutants and their corresponding phenotypes.

4. Through the manuscript, different Statistics analysis were used. Why? Please keep it consistent. For example, Line 836-840. Both one-way ANOVA, Tukey post-hoc test and Brown-Forsythe and Welch ANOVA test, Dunnett’s T3 post-hoc test were used in one panel figure.

5. Figure 5, 35s:CEP14 plants showed enhance resistance to PtoCOR-, whether this phenotype depends on CEPR2?

6. The authors' inability to detect CEP14 peptides in apoplastic wash fluids (AWFs) is a significant discrepancy that needs to be thoroughly addressed. This finding raises questions about the localization and mode of action of CEP14 and potentially weakens the overall conclusions. The authors must provide further experimental evidence or compelling explanations to resolve this issue.

**Part III – Minor Issues: Editorial and Data Presentation Modifications**

Reviewer #1: /

Reviewer #2: 1.The authors examined the requirement of hydrolation in the two proline residues of CEP14 to regulate immunity. Does the hydroxylation show selectivity in planta at these two tested sites ?

2.Fig 3A. The labeling of AEQ plants is not consistent in 3A and 3B. To rule out the dependency of CEPR1 in CEP14-induced immunity, the authors need to test the Ca2+ influx in cepr2/rlk7 to see whether the Ca2+ in completely block.

3.Figure 5B Over-expression of CEP15 could enhance plant resistance to Ptocor-, did the authors try the protection assay by infiltrate CEP15 peptide followed by the infection.

4.What is the root phenotype in these CEP OE plants? The authors performed the growth inhibition by CEP14 treatment as the marker for immune activation, given that CEPs regulate root growth and architecture, will this growth inhibition be a complex output of growth and defense pathways?

Reviewer #3: Minor comments:

1. "Throughout the manuscript (e.g., lines 1, 17, 23, 45, 75, 79), the term 'cell surface immunity' is used. While the initial ligand-receptor recognition indeed occurs at the cell surface, the subsequent immune signaling cascade takes place within the cytosol and nucleus. To more accurately reflect this, consider rephrasing 'cell surface immunity' to 'cell surface receptor-mediated immunity' or similar

2. Line 869 pv. should not be italic.

3. The presentation of data in Figure 3A and 3B needs improvement to enhance readability. The current color scheme and scaling make it difficult to discern the differences between the genotypes being compared.

4. Please provide densitometry analysis or other quantification methods to assess the levels of phosphorylated MAPK in all related figures. This quantification is essential for proper interpretation of the data.

PLOS authors have the option to publish the peer review history of their article (what does this mean? ). If published, this will include your full peer review and any attached files.

**Do you want your identity to be public for this peer review?** For information about this choice, including consent withdrawal, please see our Privacy Policy .

Reviewer #1: No

Reviewer #2: No

Reviewer #3: No

**Figure resubmission:**

**Reproducibility:**



---

## [Decision Letter · Decision Letter 1]

3 Aug 2025

PPATHOGENS-D-25-00856R1

CEPR2 perceives group II CEPs to regulate cell surface receptor-mediated immunity in Arabidopsis

PLOS Pathogens

Dear Dr. Stegmann,

Thank you for submitting the revised version of your manuscript to PLOS Pathogens. After evaluation by the previous three reviewers, we find that while progress has been made, two of the reviewers still have concerns. In particular, one reviewer feels that the previous responses did not adequately address their comments. Therefore, we are requesting a minor revision and kindly request that you provide a detailed response to the reviewers' comments. If you believe certain points raised by the reviewers are unnecessary, please provide a clear and thorough explanation to justify your position.

Please submit your revised manuscript within 30 days Oct 02 2025 11:59PM. If you will need more time than this to complete your revisions, please reply to this message or contact the journal office at plospathogens@plos.org. Please include the following items when submitting your revised manuscript:

We look forward to receiving your revised manuscript.

Kind regards,

Huiquan Liu, Ph.D.

Guest Editor

PLOS Pathogens

Bart Thomma

Section Editor

PLOS Pathogens

Sumita Bhaduri-McIntosh

Editor-in-Chief

PLOS Pathogens

orcid.org/0000-0003-2946-9497

Michael Malim

Editor-in-Chief

PLOS Pathogens

orcid.org/0000-0002-7699-2064

**Journal Requirements:**

1) In the online submission form, you indicated that  All newly generated mutant lines are available upon request to M.S. Source data are provided with this paper.. All PLOS journals now require all data underlying the findings described in their manuscript to be freely available to other researchers, either

1. In a public repository

2. Within the manuscript itself

3. Uploaded as supplementary information.

**Reviewers' Comments:**

Reviewer's Responses to Questions

**Part I - Summary**

Reviewer #1: (No Response)

Reviewer #2: This revised manuscript has been improved but there are several issues need to be addressed before accepted.

Reviewer #3: The authors have revised their conclusions in response to reviewers' comments, either by rephrasing their original statements or incorporating references to prior studies. However, these modifications appear to have inadvertently diminished the novelty of their key findings and diluted some of their initial conclusions. Proposed experiments should be performed to support their conclusions.

**Part II – Major Issues: Key Experiments Required for Acceptance**

Reviewer #1: (No Response)

Reviewer #2: (No Response)

Reviewer #3: (No Response)

**Part III – Minor Issues: Editorial and Data Presentation Modifications**

Reviewer #1: (No Response)

Reviewer #2: 1.The author claimed the response is fully abolished in cepr2 rlk7 double mutants in Fig. S3D, but I did not find the Fig. S3D in the new version.

2.In Figure 3A, the Ca2+ influx is comparable in cepr2 rlk7 and cepr2, I do not think it is completely blocked in cepr2 rlk7 as the authors explained. It should be described more accurate.

3.I did not get the reason why the authors do not test the protection assay. The Ca2+ flux and MAPK activation they tested are all CEPs-induced immunity.

4.“We observed that both CEP13 overexpression lines displayed reduced primary root growth, while only one of three lines each for 35S::CEP14 and 35S::CEP15 displayed this phenotype. We included this data in the new Supplementary Fig. 5E.” The FigS5E is not root phenotype data.

Reviewer #3: (No Response)

PLOS authors have the option to publish the peer review history of their article (what does this mean? ). If published, this will include your full peer review and any attached files.

**Do you want your identity to be public for this peer review?** For information about this choice, including consent withdrawal, please see our Privacy Policy .

Reviewer #1: No

Reviewer #2: No

Reviewer #3: No

**Figure resubmission:**
---

## [Editor Report · Decision Letter 2]

30 Aug 2025

Dear Dr Stegmann,

We are pleased to inform you that your manuscript 'CEPR2 perceives group II CEPs to regulate cell surface receptor-mediated immunity in Arabidopsis' has been provisionally accepted for publication in PLOS Pathogens.

Best regards,

Huiquan Liu, Ph.D.

Guest Editor

PLOS Pathogens

Bart Thomma

Section Editor

PLOS Pathogens

Sumita Bhaduri-McIntosh

Editor-in-Chief

PLOS Pathogens

orcid.org/0000-0003-2946-9497

Michael Malim

Editor-in-Chief

PLOS Pathogens

orcid.org/0000-0002-7699-2064
---

## [Editor Report · Acceptance letter]

Dear Prof. Dr. Stegmann,

We are delighted to inform you that your manuscript, " 

CEPR2 perceives group II CEPs to regulate cell surface receptor-mediated immunity in Arabidopsis," has been formally accepted for publication in PLOS Pathogens.

Best regards,

Sumita Bhaduri-McIntosh

Editor-in-Chief

PLOS Pathogens

orcid.org/0000-0003-2946-9497

Michael Malim

Editor-in-Chief

PLOS Pathogens

orcid.org/0000-0002-7699-2064